# Prediction of Cognitive Degeneration in Parkinson’s Disease Patients Using a Machine Learning Method

**DOI:** 10.3390/brainsci12081048

**Published:** 2022-08-07

**Authors:** Pei-Hao Chen, Ting-Yi Hou, Fang-Yu Cheng, Jin-Siang Shaw

**Affiliations:** 1Department of Neurology, MacKay Memorial Hospital, Taipei 104217, Taiwan; 2Institute of Long-Term Care, Mackay Medical College, New Taipei City 252, Taiwan; 3Institute of Mechatronic Engineering, National Taipei University of Technology, Taipei 106344, Taiwan

**Keywords:** Parkinson’s disease, machine learning, neuropsychological test, biomarker

## Abstract

This study developed a predictive model for cognitive degeneration in patients with Parkinson’s disease (PD) using a machine learning method. The clinical data, plasma biomarkers, and neuropsychological test results of patients with PD were collected and utilized as model predictors. Machine learning methods comprising support vector machines (SVMs) and principal component analysis (PCA) were applied to obtain a cognitive classification model. Using 32 comprehensive predictive parameters, the PCA-SVM classifier reached 92.3% accuracy and 0.929 area under the receiver operating characteristic curve (AUC). Furthermore, the accuracy could be increased to 100% and the AUC to 1.0 in a PCA-SVM model using only 13 carefully chosen features.

## 1. Introduction

Parkinson’s disease (PD) is a neurodegenerative disease. Clinical motor dysfunctions, such as resting tremors, rigidity, bradykinesia, postural instability, and inability to initiate motion, are commonly seen in patients with PD. In addition to motor dysfunction, patients with PD also tend to have cognitive impairments, such as mild cognitive impairment (MCI) and dementia. MCI and dementia may also affect motor dysfunction in PD patients, and there is a complicated relationship between motor function and cognition in patients with PD [1].

According to previous research, there is a high probability that patients with PD develop cognitive impairment that may affect their quality of life; this impairment predominantly involves the cognitive domains of attention, executive function, and visuospatial skills [2,3,4]. Biomarkers obtained mainly from neuroimaging data were extensively discussed for finding predictors of cognitive dysfunction in Parkinson’s disease in a literature survey [5,6]. Indeed, it is crucial to identify the factors influencing cognitive decline that affect clinical prognosis and require early intervention [7].

Machine learning in artificial intelligence is popular in constructing a predictive model. In a study with 45 subjects, four machine learning models were developed to assess the ability to discriminate between PD patients with cognitive integrity (PDCI), mild cognitive impairment (PDMCI), and dementia (PDD). In an SVM model for classifying PDD and PDCI, the most relevant variables related to PD dementia were white matter, lateral ventricle, and hippocampus volume, and the prediction accuracy could reach 96.67% [8]. In another study with a cohort of 75 PD patients, a set of five biomarkers (cerebrospinal fluid (CSF) total tau levels, CSF phosphorylated tau levels, CSF Aβ42 levels, APOE genotype, and SPARE-AD imaging score) was adopted as the predictor of a logistic regression classifier, and 80% accuracy was achieved in discriminating PD patients with normal cognition from PD patients with dementia [9].

In this preliminary study, a cross-sectional investigation of clinical variables, neuropsychological test results, and plasma biomarkers [10,11,12] in patients with PD was conducted to identify features related to cognitive impairment. More specifically, machine learning was applied to obtain a predictive cognitive degeneration model and ascertain key predictors that help medical experts quickly identify a patient’s cognitive condition and provide treatment.

## 2. Methods

### 2.1. Participants

This cross-sectional study recruited patients with PD from October 2019 to November 2019 and from July 2020 to November 2020. The patients were recruited at the Neurology Department of the MacKay Memorial Hospital (Taiwan).

The study was performed following the Declaration of Helsinki and was approved by the Institutional Review Board of Mackay Memorial Hospital in Taiwan (IRB Number: 18MMHIS152). Informed consent was obtained from all participants. A consecutive series of patients with PD were recruited in the Neurology outpatient clinics of a tertiary medical center in northern Taiwan from October 2019 to November 2020. All participants met the following criteria: (a) age > 30 years, (b) diagnosed with idiopathic PD according to the PD clinical diagnostic criteria of the Movement Disorder Association [13,14], and (c) no diagnosis of dementia (for those who have received more than six years of education, the Mini-Mental State Evaluation [MMSE] score must be >23 points; for those who have less than six years of education, the MMSE must be >13 points). Participants were excluded if (a) they had more than two incomplete tests or (b) uncontrolled medical conditions that cause severe physical and cognitive disabilities. A physician evaluated the presence of the exclusion criteria, and the process was shown in Figure 1.

### 2.2. Clinical Data

We collected clinical information from patients, including sex, age, course of the disease, education level, levodopa dose, Barthel index, Hoehn and Yahr stage, and Unified Parkinson’s Disease Rating Scale (UPDRS) parts I–III subscale scores [15,16,17,18,19]. 

Trained nurses performed a comprehensive neuropsychological assessment of all patients. The assessment includes general cognition and specific cognitive domains involving the following examinations: (1) global cognition (MMSE and Clinical Dementia Rating-Sum of Boxes [CDR-SB]); (2) processing speed and working memory (Digits Recall Forward and Backward); (3) verbal learning and memory (California Verbal Language Test-II Short Form [CVLT-SF]); (4) semantic verbal fluency (animal naming); (5) language (Boston Naming Test); (6) attention and visuospatial processing (Trail Making Test A and B [TMT-A and TMT-B]); and (7) visuoperceptual and visuospatial processing (Benton Judgement of Line Orientation) [20,21,22,23,24,25,26,27].

### 2.3. Neurobiological Indicator

A blood sample of 10 mL was collected from each subject and centrifuged within one hour of collection. The plasma was separated and immediately frozen in test tubes at −80 °C. We then delivered frozen plasma on dry ice to MagQu Co., Ltd. (New Taipei City, Taiwan) and measured the levels of plasma α-syn, Aβ42, and t-tau using an immunomagnetic reduction assay.

### 2.4. Data Analysis

This study collected 29 clinical data and the three plasma biomarkers for each participant, as shown in Table 1. In addition, 42 patients with these complete data were included to build a classification model using support vector machine (SVM) and principal component analysis (PCA) in the Python Sklearn package. It was previously shown that the PCA–SVM method effectively classified PD–MCI from non-PD–MCI patients with high accuracy, provided good predictors were used [28].

### 2.5. Data Normalization

Before using the SVM prediction model, it is necessary to preprocess the collected data to obtain a better data structure for training and avoid differences in the data distribution area, which affects the convergence speed and accuracy of the prediction model. Normalization is a standard preprocessing technique [29]. Min-max normalization is used in the data preprocessing. The data are scaled to between 0 and 1 through normalization without changing the distribution of the data [30] using the following transformation:(1)$$x_{norm}=\frac{x-x_{min}}{x_{max}-x_{min}}$$
where $x_{max}$ is the maximum value, $x_{min}$ is the minimum value, and $x_{norm}$ is the normalized value between 0 and 1 for the dataset, *x*.

### 2.6. SVM

SVMs are a type of supervised learning method. It is to find a hyperplane between two-class categories. The SVMs try to find the decision boundary in the training data set to maximize the margin between the two classes to reduce the generalization error of the classifier. The maximum boundary hyperplane can be determined through various kernels to build a linear or nonlinear classification [31,32,33]. This study uses different kernel functions, including linear, RBF (radial basis function), and Poly (polynomial) functions to compare which model is better for predicting cognitive impairment.

### 2.7. PCA

PCA is an unsupervised learning method for feature extraction. Using the first few principal components (PCs) of the covariance matrix, normalized high-dimensional data can be projected into a lower-dimension space using orthogonal transformation while preserving the essential features [34,35,36]. More specifically, the dimensionality of the original dataset X ∈ $R^{n\times p}$ (i.e., *n* samples and *p* features) can be reduced to X′ ∈ $R^{n\times s}$ by PCA with *s* < *p*. That is, X′ with less dimension presents the data more concisely while retaining most of the key features (the cumulative energy of the first s eigenvalues of the covariance matrix is above a certain threshold, for example, 90%, of the total energy). The new features are then provided to the SVM with a lower dimension for predictive classification; hence, the training model can accelerate the calculation and improve the accuracy.

### 2.8. Area under the Receiver Operating Curve

The receiver operating characteristic curve (ROC curve) is drawn as a plot with the false positive rate (FPR) as the X-axis, and the true positive rate (TPR) as the Y-axis that illustrates the diagnostic ability of a classifier as its discrimination threshold is varied. The area under the ROC curve (AUC) measures the power of a classifier to distinguish between classes and is used as a summary of the ROC curve [37]. The higher the AUC, the better the model’s performance at distinguishing between the positive and negative classes. When the AUC is equal to 0.5, the classifier cannot differentiate between positive and negative categories. Therefore, an AUC between 0.9 and 1 indicates that the predictive classifier has an excellent discriminatory ability.

## 3. Results

After one year of data collection, there were 116 patients with idiopathic PD. Of those, 41 patients refused blood and neuropsychological tests, five were transferred to another hospital, six lost contact, and 22 had incomplete data. Ultimately, only 42 patients had complete data. In this study, we used CDR-SB scores for the two classifications. The score interval for the patients without cognitive impairment (from normal to MCI) was ≤0.5, and the score interval for those with moderate and severe cognitive impairment was >0.5. After judging and categorizing, 16 patients were classified as not having cognitive impairment, and 26 patients had moderate to severe cognitive impairment. The demographic and collected data for these two groups were presented in Table 2.

It was worth mentioning why CDR-SB scores were used for the dichotomy of cognitive degeneration. A more quantitative representation of the CDR is provided by the sum of the severity ratings for the six cognitive and functional domains. CDR-SB provides a more quantitative measure of dementia severity than the global CDR. The CDR-SB frequently assesses Alzheimer’s disease progression in clinical research [38,39] and has been used in patients with Parkinson’s disease [40]. Owing to the increased range of values, the CDR-SB offers several advantages over the global score, including increased utility in tracking changes within and between stages of dementia severity. Unlike the other global cognitive testing (i.e., MMSE) in this study, CDR is not influenced by age, education, and gender.

First, all variables (*p* = 32) were included as feature inputs; 70% of the 42 patients were randomly selected as the training set and 30% as the verification set. Different kernel functions were used to train the SVM and PCA–SVM classification models. The validation accuracy under the full-parameter linear function reached 84.6%, and the AUC was 92.9%. After reducing the dimensionality of the original 32 features using PCA to six features, the accuracy increased to 92.3% for the same AUC rate. After PCA’s dimensionality reduction, the overall forecast confidence improved, as shown in Table 3 and Figure 2.

Second, from the above results, it suggested that a set of more concise predictors was possible. The six items (Hoehn–Yahr stage, IADL, Barthel Index, UPDRS I, II, and III) are related to essential motor and non-motor functions in PD patients. They are commonly used as clinical tools to assess PD patients. For the advanced neuropsychological tests, we selected four tests on executive functioning (TMT-B, Verbal fluency, Digits Forwards and Backwards) based on previous research [41,42,43] showing that executive dysfunction was joint in PD, especially early PD. The three biomarkers (α-syn, Aβ42, and t-tau), typically pathognomonic for the pathology of PD and AD, were also included which could predict executive dysfunction and cognitive decline in PD [12,44]. Therefore, a total of condensed 13 parameters were chosen as feature inputs as shown in Table 4. A randomly selected set of 70% of the 42 patients was used to train the prediction model, and the remaining 30% were used to verify the model performance. Different kernel functions were used to train the SVM and PCA–SVM classification models. The validation accuracy under the linear function in the SVM classification model reached 84.6%, and the AUC reached 100%. When reducing the dimensionality of the 13 features using PCA to three features, the accuracy under the linear function significantly improved to 100%. The AUC was maintained at 100%, as shown in Table 5 and Figure 3.

## 4. Discussion

In this study, machine learning was used to accurately classify the presence or absence of cognitive disorders in terms of CDR-SB scores in patients with idiopathic PD. In particular, we selected ten parameters related to clinical data and dynamic execution in neuropsychological tests and the three plasma biological indicators shown in Table 4 as the predictors that led to an accuracy rate and AUC for the PCA-SVM model as high as 100%. Therefore, dynamic execution and plasma biometrics are highly relevant for assessing the cognitive ability of PD patients. Compared to the two previously mentioned machine learning models for predicting cognitive degeneration [8,9], the developed PCA-SVM model produced the best prediction accuracy. In addition, literature on the use of the standard clinical assessment tools including neuropsychological tests for PD patients as cognitive predictors in a machine learning was limited and was even rarely seen using plasma biomarkers.

CDR is generally used to determine the severity of a patient’s overall cognitive status, which is time-consuming and requires professional judgment. A patient’s cognitive ability cannot be determined by questionnaires alone. However, only ten questionnaire items from clinical and neuropsychological tests and three plasma biological indicators were needed to train the predictive model through this training model. It is noted that questionnaires can be readily implemented after suitable personnel training and do not necessarily require professional medical persons; thus, it can reduce the time and burden on medical persons.

This study has some limitations. First, this was a cross-sectional study. Longitudinal studies are needed to identify the key indicators that can predict cognitive degeneration in a future time in PD patients, trace these predictors in the different disease stages, and clarify their roles in other cognition domains. Second, the sample size of this study was relatively small because of the need for neuropsychological evaluations and blood tests. Third, there was a lack of a control group of healthy subjects to compare the levels of these indicators. Finally, although all our participants fulfilled the diagnostic criteria of clinically established or probable PD, the possibility of overlapping clinical manifestation and misdiagnosis of progressive supranuclear palsy-parkinsonism predominant type (PSP-P) and postural instability and gait difficulty subtype of PD should be emphasized [45].

For future perspectives, increasing the sample size and conducting a longitudinal study are among the priorities. Specifically, increasing the sample size can further validate and support the developed model’s performance in view of the small sample size in this study. Conducting longitudinal study can identify the key indicators and help develop a prediction model that can predict cognitive degeneration in a future time in PD patients, which is extremely important for medical experts to quickly identify a patient’s cognitive condition and provide treatment in advance.

## Figures and Tables

**Figure 1 brainsci-12-01048-f001:**
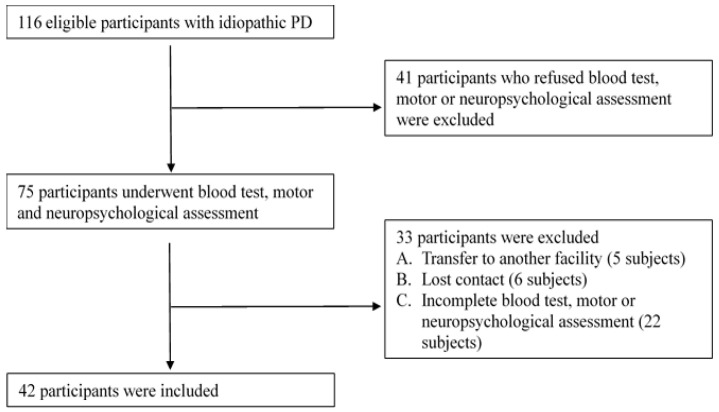
Exclusion flow chart.

**Figure 2 brainsci-12-01048-f002:**
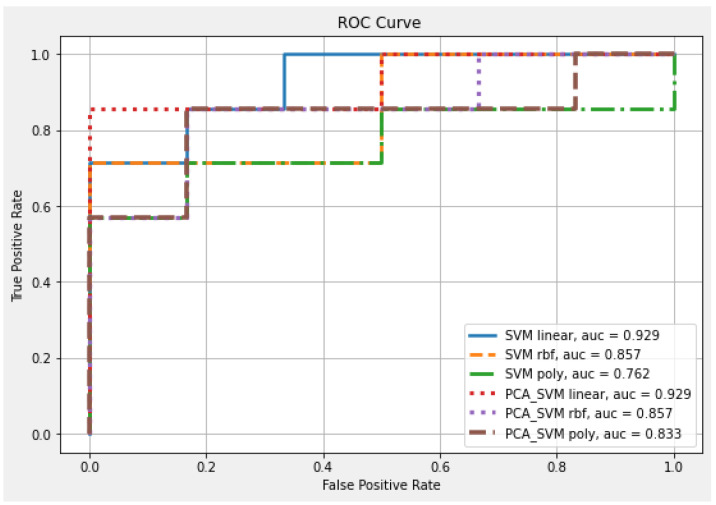
ROC curve and AUC results for each 32-parameter classifier of CDR-SB deterioration.

**Figure 3 brainsci-12-01048-f003:**
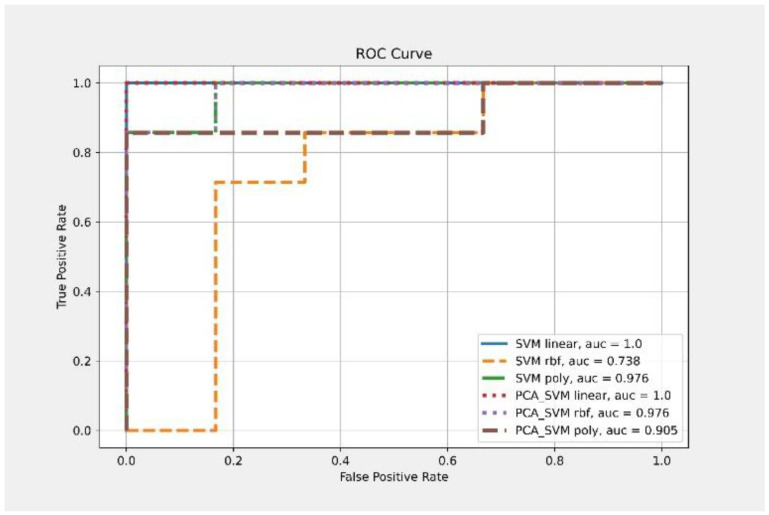
ROC curve and AUC result for each 13-parameter classifier of CDR-SB deterioration.

**Table 1 brainsci-12-01048-t001:** Twenty-nine clinical data and the three plasma biomarkers.

Hoehn–Yahr Stage	UPDRS I	UPDRS II	UPDRS III
LED (mg/day)	Gender	Age of visits	Age of onset
Disease duration	Education (years)	Barthel Index	MMSE
IADL	JLO	PSQI	EQ-5D index
EQ-5D VAS	GDS−15	GAD−7	TMT-A
TMT-B	Verbal fluency	Digits Forwards	Digits Backwards
CVLT-SF total recall	CVLT-SF Immediate	CVLT-SF delay	CVLT-SF recognition
BNT	α-syn (pg/mL)	Aβ42 (pg/mL)	t-tau (pg/mL)

Abbreviation: Aβ42, amyloid-β 42; BNT, Boston Naming Test; CVLT-SF, California Verbal Learning Test-Short Form; EQ-5D, EuroQol-5 dimensions; GAD-7, Generalized anxiety disorder scale 7-item; GDS-15, Geriatric depression scale 15-item; IADL, Instrumental activities of daily living; JLO, Judgment of Line Orientation; LED, Levodopa equivalent dose; MMSE, Mini-Mental State Examination; PSQI, Pittsburgh sleep quality index; SD, Standard Deviation; TMT, Trail Making Test; UPDRS, Unified Parkinson’s Disease Rating Scale; VAS, visual analog scale; t-tau, total tau; α-syn, α-synuclein.

**Table 2 brainsci-12-01048-t002:** The demographic and data comparisons of the participants.

*N* = 42	Without Cognitive Impairment (*N* = 16)	Moderate and Severe Cognitive Impairment (*N* = 26)	*p* Value
Hoehn–Yahr stage	1.78 (0.73)	2.37 (0.61)	0.291
UPDRS I	2.38 (1.147)	4.15 (1.78)	0.078
UPDRS II	5.63 (2.391)	11.23 (5.88)	0.002
UPDRS III	12.63 (5.35)	20.65 (10.35)	0.013
LED (mg/day)	428.56 (229.13)	440.77 (241.8)	0.617
Gender	Male 8/50%	Male 10/38.46%	0.463
Age of visits	68.38 (8.57)	76.65 (7.27)	0.417
Age of onset	65.81 (8.72)	71.92 (8.19)	0.753
Disease duration	2.56 (2.39)	4.73 (3.52)	0.022
Education (years)	7.69 (3.22)	7.04 (4.96)	0.114
Barthel Index	156.25 (225)	88.27 (16.31)	0.019
MMSE	26.94 (2.24)	22.96 (3.96)	0.015
IADL	23.38 (1.26)	17.38 (6.76)	0.000
JLO	14.5 (4)	12.23 (4.86)	0.366
PSQI	5.38 (2.39)	7 (2.79)	0.71
EQ-5D index	0.77 (0.17)	0.75 (0.21)	0.78
EQ-5D VAS	68.88 (10.78)	66.54 (16.54)	0.335
GDS−15	2.5 (3.16)	3.54 (4.71)	0.067
GAD−7	1 (1.86)	2.08 (3.5)	0.068
TMT-A	27.19 (10.88)	36.62 (12.74)	0.494
TMT-B	72.06 (28.19)	87.96 (33.74)	0.15
Verbal fluency	11.56 (4.56)	9.27 (3.76)	0.426
Digits Forwards	7.38 (1.31)	6.12 (1.58)	0.21
Digits Backwards	5.19 (1.56)	3.58 (1.53)	0.897
CVLT-SF total recall	19.94 (5.89)	17.54 (4.42)	0.440
CVLT-SF immediate	6 (1.75)	4.96 (1.8)	0.784
CVLT-SF delay	4.69 (2.06)	3.81 (1.96)	0.696
CVLT-SF recognition	5.69 (2.44)	4.65 (2.45)	0.461
BNT	23.88 (2.99)	19.08 (6.46)	0.006
α-syn (pg/mL)	0.1 (0.05)	0.12 (0.05)	0.793
Aβ42 (pg/mL)	16.66 (0.45)	16.7 (0.59)	0.669
t-tau (pg/mL)	22.75 (2.63)	23.62 (3.63)	0.162

**Table 3 brainsci-12-01048-t003:** Thirty-two parameter set to predict CDR-SB deterioration.

Classifier	Kernel	Feature Number	Accuracy	AUC
SVM	Linear	32	0.846	0.929
	RBF		0.769	0.857
	Poly		0.615	0.762
PCA-SVM	Linear	6	0.923	0.929
	RBF		0.769	0.857
	Poly		0.615	0.833

**Table 4 brainsci-12-01048-t004:** Condensed thirteen parameters as the model predictors.

Hoehn–Yahr Stage	IADL	Barthel Index
UPDRS I	UPDRS II	UPDRS III
Verbal fluency	Digits Forwards	Digits Backwards
TMT-B	α-syn	Aβ42
t-tau		

**Table 5 brainsci-12-01048-t005:** Thirteen selected parameters to predict CDR-SB deterioration.

Classifier	Kernel	Feature Number	Accuracy	AUC
SVM	Linear	13	0.846	1
	RBF		0.538	0.738
	Poly		0.846	0.976
PCA-SVM	Linear	3	1	1
	RBF		0.923	0.976
	Poly		0.692	0.905

## Data Availability

The data that support the findings of this study are available on request from the corresponding author. The data are not publicly available due to the privacy of participants.

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
