# Peer review of "Prediction of Cognitive Degeneration in Parkinson’s Disease Patients Using a Machine Learning Method"

_brainsci, 2022, doi:10.3390/brainsci12081048_

Round 1

Reviewer 1 Report

The cognitive decline in Parkinson's disease is a striking issue requiring further analysis. Authors elaborate on the prediction of this process using machine learning method. I have several points, which should additionally be addressed:

1. In the introduction authors mention features of PD, however all of the mentioned factors can be observed in PSP-P. The overlapping of clinical manifestation and possible misdiagnosis should be additionally emphasized. Ref.

Progressive Supranuclear Palsy-Parkinsonism Predominant (PSP-P)-A Clinical Challenge at the Boundaries of PSP and Parkinson's Disease (PD). Front Neurol. 2020 Mar 10;11:180. doi: 10.3389/fneur.2020.00180. PMID: 32218768; PMCID: PMC7078665.

2. Authors should additionally discuss the methodological limitations of the study. The part of the discussion regarding this issue is too vague.

3. It would be valuable to elaborate on future perspectives.

Reviewer 2 Report

Thanks for recommending me as a reviewer. In this paper, authors developed a predictive model for cognitive degeneration in patients with Parkinson's disease using a machine learning method. In this paper, the clinical data, plasma biomarkers, and neuropsychological test results of patients with PD were collected and utilized as model predictors. In this study, machine learning methods comprising support vector machines and principal component analysis  were applied to obtain a cognitive classification model. If authors complete revisions, the quality of the study will be further improved.

1. The introducution section is well written. The quality of the research will be improved if the authors describe in more detail the trends of previous research related to prediction of cognitive degeneration in Parkinson's disease using a machine learning in the introduction section.

2. line 50-56: Authors should describe in more detail the criteria for selection and exclusion of subjects in the Methods section.

3. Is Table 1 necessary for this study?

4. In Table 3, the authors give significance levels. There is no need for a separate footnote on significance level.

5. In the discussion section, authors should interpret the main findings of the study in comparison with previous studies.

6. Authors should add limitations to the discussion section.

Reviewer 3 Report

1. Introduction part is too condensed to fully understand the current status of the topic of this study.

2. Please provide a current situation about "patients with PD have not been fully discussed" in detail.

3. The authors should present the findings of relevant studies focusing on machine learning related to predicting cognitive declines in PD in the last part of the Introduction

4. Compared to conventional statistics, what could machine learning bring advantages?

5. In the Method section, please provide the reason why subjects aged over 30 years were recruited.

6. Sample size is too small for machine learning. Please provide relevant studies including a small sample size of less than 50 for machine learning.

7. Even though the previous study[24] reported that the SVM and PCA were quite good, the authors need to present the appropriate reason why only two machine learning methods were chosen as the cited study seems to be related to gait-baed machine learning.

8. In the Result section, how were 13 parameters selected? Were there any statistical criteria?

9. Discussion is too short to fully cover relevant research. Specifically, there is no comparison with relevant studies. In addition, the authors should provide how a very high accuracy could be accomplishmented.

Round 2

Reviewer 1 Report

I do not have further comments.

Reviewer 2 Report

The authors have faithfully completed the revision.

Reviewer 3 Report

Even though, some points seemed to be appropriately addressed, a major concern about the discussion which could not still cover relevant studies remains. Specifically, only three prior studies were compared to this study.